# AI Ethics—A Bird's Eye View

**Maria Christoforaki** * and **Oya Beyan**

Institute for Medical Informatics, University Hospital Cologne, Faculty of Medicine, University of Cologne, Kerpener Strasse 62, 50937 Cologne, Germany; oya.beyan@uk-koeln.de
* Correspondence: maria.christoforaki@uk-koeln.de

**Abstract:** The explosion of data-driven applications using Artificial Intelligence (AI) in recent years has given rise to a variety of ethical issues regarding data collection, annotation, and processing using mostly opaque algorithms, as well as the interpretation and employment of the results of the AI pipeline. The ubiquity of AI applications negatively impacts a variety of sensitive areas, ranging from discrimination against vulnerable populations to privacy invasion and the environmental cost that these algorithms entail, and puts into focus on the ever present domain of AI ethics. In this review article we present a bird's eye view approach of the AI ethics landscape, starting from a historical point of view, examining the moral issues that were introduced by big datasets and the application of non-symbolic AI algorithms, the normative approaches (principles and guidelines) to these issues and the ensuing criticism, as well as the actualization of these principles within the proposed frameworks. Subsequently, we focus on the concept of responsibility, both as personal responsibility of the AI practitioners and sustainability, meaning the promotion of beneficence for both the society and the domain, and the role of professional certification and education in averting unethical choices. Finally, we conclude with indicating the multidisciplinary nature of AI ethics and suggesting future challenges.

**Keywords:** AI ethics; ethics guidelines; data science; artificial intelligence; responsibility; education





## 1. Introduction

The explosion of data-driven applications using Artificial Intelligence (AI) in recent years has given rise to a variety of ethical issues that range from discrimination against vulnerable populations to privacy invasion [1] and the environmental impact of these algorithms [2], putting into focus the ever present domain of AI ethics. The call for regulation was answered by professional, national, and international organizations, most recently by the European Union as well [3].

In this paper we make a brief introduction of AI ethics, starting from a historical perspective and sketch today's landscape with broad strokes. We also extended to broader concerns, such as assigning responsibility and accountability for the use of AI both in an individual as well as a societal level. We see AI ethics as a multidisciplinary domain that has philosophical, legal, societal, and technological aspects and aim to present these different facets and their interactions. Taking into consideration all these dimensions, offers us a deeper understanding and enables us to face more effectively the challenges rising from the ubiquitous introduction of AI, especially in areas such as healthcare.

While we use the term AI in the rest of this paper, we will include sources that deal not only with AI but also Machine Learning (ML) and Data Science (the last one we treat as synonymous with Big Data).

It is not uncommon to find combinations of the three terms in research publications, AI being the oldest one (indicatively, using Google Scholar at 3 July 2021 to search the terms AI, Big Data, Data Science, and Machine Learning yielded the following: 'Big Data and Machine Learning' 3140,000 articles; 'AI and Big Data' 2,950,000 articles; 'ai and data science' 5,040,000 articles; 'AI and Machine Learning' 2,380,000 articles; and 'AI Machine Learning

data science' 1,640,000 articles). Dating as far back as 1956 [4] (p. 1), the term faded for a long time during the so called *AI Winter*, a term that was coined in the 1980s to describe a period when interest and funding for AI was minimal [5] (p. 408), to be revitalized in recent years by the introduction of data-driven algorithms, notably Deep Learning (DL). DL being the rebranding of Artificial Neural Networks, the prototype of which was introduced in 1943 and achieved fame for the first time in the 1990s [6,7]. Machine Learning (ML) is considered part of AI and according to Mitchell "is concerned with the question of how to construct computer programs that automatically improve with experience" [8] (p. xv).

De Mauro, Greco, and Grimaldi define Big Data as "the Information asset characterized by such a High Volume, Velocity and Variety to require specific Technology and Analytical Methods for its transformation into Value" [9] (p. 122). boyd and Crawford [10] (p. 663) give a definition for Big Data that could hold also for AI and ML and gives a more encompassing and multidisciplinary aspect that underlines also the perspective of this paper:

"We define Big Data as a cultural, technological, and scholarly phenomenon that rests on the interplay of:

(1) Technology: maximizing computation power and algorithmic accuracy to gather, analyze, link, and compare large data sets.

(2) Analysis: drawing on large data sets to identify patterns in order to make economic, social, technical, and legal claims.

(3) Mythology: the widespread belief that large data sets offer a higher form of intelligence and knowledge that can generate insights that were previously impossible, with the aura of truth, objectivity, and accuracy."

Taking all the above into consideration, in this paper we are interested in the ethical issues that relate to AI, as it evolved (mainly) in the last decade and is characterized by the development of data-driven, self-learning, intelligent systems that mimic or supplant human decision-making, and involve collecting, processing, and analyzing big datasets, using not always transparent algorithms with disparate social impact.

The rest of the paper is structured as follows: Section 2 presents the definitions and the accompanying concepts regarding AI ethics, Section 3 covers the issues that are related to the introduction of Big Data sets, Section 4 covers the guidelines that were developed for the ethical use of AI and the accompanying problematic, Section 5 examines how these principles are brought into action within specific frameworks, Section 6 discusses how to assign responsibility and accountability regarding the use of AI systems, while in Section 7 we offer conclusions and some insight on future issues

## 2. AI, Computer, Machine and Computational Ethics

AI ethics is defined by Leslie as "a set of values, principles, and techniques that employ widely accepted standards of right and wrong to guide moral conduct in the development and use of AI technologies" [11] (p. 3).

AI ethics is related to computer, machine, and computational ethics. While computer ethics concentrates on the ethical issues that arise by the use of computers by humans [12], in machine ethics, the machine can be regarded as an ethical actor to the extent that it can make decisions that have ethical consequences regarding human users, and perhaps other machines [13]. Machine ethics is regarded as an extension of computer ethics [14], while Segun examines machine ethics as a subarea of the ethics of AI [15]. However, whether the concept covers all of AI ethics or is a part of it is not clearly defined [16].

Computational ethics is concerned with making ethics computable, insofar AI systems, for example robots or autonomous systems, can make decisions that may have ethical consequences [15]. Whether ethics is computable is addressed by Moor [17], the main two questions being: (a) if a machine (or piece of software in this case) can be regarded as a moral agent, and (b) how a moral judgment is achieved. In this context, the term of artificial moral agent (AMA) is used to characterize a piece of software or a robot that is capable of engaging in moral behavior or at least of avoiding immoral behavior, which may be determined by some ethical theory [18]. AMAs are distinguished by Moor into

ethical-impact, implicit, explicit, and full ethical agents. Ethical impact agents can have praiseworthy or blameworthy impact by their mere existence and use (or misuse), for example by replacing a person in a job with a machine or giving an advantage to the person that is using it. Implicit ethical agents support ethical behavior or avoid unethical behavior by the way they are constructed, for example when a program enforces security and safety constraints. Explicit ethical agents have explicitly stated ethical rules and are able to make ethical judgments by applying some form of calculation. Full ethical agents can make ethical decisions and justify them, for example adult humans having consciousness, intentionality, and free will in [19].

In order to calculate an ethical decision, three approaches can be followed: top-down where the moral judgment is determined by a philosophical approach; bottom-up, where moral behavior is identified and rewarded so that the system is trained to distinguish between right and wrong (essentially a machine learning approach); and finally, a hybrid that combines both [20].

Moral judgments can be calculated using a variety of philosophical approaches mainly consequentialist and (e.g., utilitarianism) and deontological theories. According to consequentialism "[a]n act is right or wrong according to its consequences[and] is right if and only if it is reasonably expected to produce the greatest good or least harm in comparison with alternative action choices" [21] (p. 46). A strong version of consequentialism is utilitarianism. Utilitarianism was introduced in the nineteenth century by Jeremy Bentham and John Stuart Mill. A modern day representative of consequentialism is Peter Singer [22]. In deontological theories right action is defined directly in terms of moral principles manifest in duties [21] (p. 51). Deontological ethics is usually associated with Immanuel Kant [21]). However, virtue ethics, particularism, and hybrid approaches [18,23], as well as discourse ethics [24] are also used.

Regarding the evaluation of artificial agent implementations, recent reviews observe that there are only a few prototypes that can deal with some moral issues [18] and there is no providence for adding cultural influences or societal preferences in existing systems. The benchmarks to evaluate existing approaches are lacking and there are no specific tasks to be implemented, no consensus as to what the correct output is, and few datasets to use in an implementation [23].

The moral status of AI systems is being debated, very often in the context of Artificial General Intelligence (AGI) as well as Artificial Super Intelligence (ASI) and the concept of Singularity AGI refers to systems "that possess a reasonable degree of self-understanding and autonomous self-control, and have the ability to solve a variety of complex problems in a variety of contexts, and to learn to solve new problems that they didn't know about at the time of their creation" [25] (p. VI), while Singularity and the related notion of ASI refer to a point "at which [AI systems] reach a superhuman level of performance" [4] (p. 12), and [16]. However, while the prospect of the latter may not seem so imminent, philosophical papers such as [26] point out issues such as responsibility, transparency, and auditability which are already at the focus of the present day discussion.

While AGI/ASI systems seem like a remote probability, for the time being at least, automated decision-making is greatly disputed. The most prominent subjects are lethal autonomous weapons systems [27], autonomous vehicles [28], and healthcare AI applications [29] (the last two areas often compared to each other as to the levels of autonomy [30], as well as the fostering of trust [31]). Specifically in healthcare, the application of AI, as pointed out by London [32], evokes strong reactions because of the possibility of delegating to a machine life-and-death decisions that are normally made by highly trained professionals. Grote and Berens [29] underline, that while the application of algorithmic decision-making in healthcare greatly enhances the capabilities of the domain professionals, especially in the area of medical diagnosis, it is also accompanied by epistemic and normative pitfalls.

These issues, however, are common to most of AI application deployment, and as such, they will be discussed in the rest of this paper. Specifically, their emergence and ramifications, as well as ways to tackle the challenges they present.

### 3. Enter the Big Datasets

#### 3.1. Data Collection and Manipulation Issues

The advent and subsequent flourish of data science and Big Data technologies, especially when coupled with AI, focused the conversation on data, their production, manipulation, sharing, safekeeping, etc. [33–36].

The mere accumulation of large datasets either via data mining or collection from users (e.g., through social media apps, private companies, governmental, and public organizations such as healthcare services) raised issues such as: privacy; confidentiality, and anonymity; unauthorized use for a different purpose than originally collected for, and given or extracted consent by the users; data ownership, control, and monetization; data accuracy and validity, namely whether the data that are being used are appropriate for the problem addressed; and trustworthiness of the algorithms and the people who handle that data [37,38].

#### 3.2. Sources of Harm: The Black-Box Problem and Bias in the AI/ML Pipeline

Additionally, the opacity of the AI algorithms created a question of trust demanding justifications for the decisions that are made by these algorithms, especially in the case where they demonstratively exhibited bias leading to discrimination [39–41]

Bias can enter in each step of an AI/ML pipeline. Suresh and Guttag [42] offer a comprehensive description framework that identifies potential sources of harm. It spans from the creation of the datasets (data collection and preparation), to model building and implementation (model development, evaluation, post-processing, and deployment). The creation of the datasets can be adversely impacted by (a) historical bias, which incorporates already existing bias in the world; (b) representational bias, i.e., underrepresentation of specific populations during the sampling process; and (c) measurement bias, where the choice of features and labels that are assigned to data that work as proxies for the desired quantities is poor. These probably already problematic datasets are fed to a model which may be impacted by (a) aggregation bias that is created by combining heterogeneous datasets during model construction; (b) evaluation bias, i.e., when the benchmark population does not correspond to real-world cases; and (c) deployment bias, where the outcomes of the system are interpreted and used in an inappropriate manner.

#### 3.3. Weapons of Math Destruction

This framework is mirrored partly to the types of ethical concerns that were raised by algorithms that were identified by Mittelstadt et al. [43]: three epistemic (having to do with the data and model deployed), namely, inconclusive, inscrutable or misguided evidence; and two normative ones, i.e., the products of the pipeline can be unfair, leading to discrimination and thus, having transformative effects on the life of individuals and/or societal impact. The final issue they set has to do with the traceability of the algorithms and responsibility assignment.

All the above vulnerabilities led to discriminative and harmful applications of AI, indicatively, racial [44] and gender discrimination, e.g., in ad targeting [45] and natural language processing (NLP) [46], geo-diversity issues [47], and abusive NLP [48], among others (for a non-academic presentation of impacted sectors see [1], from which the title of this subsection is borrowed).

In healthcare, specifically, racial discrimination is documented to exist in a variety of AI applications. Starting from appointment scheduling, black patients are set to wait about 30% longer than non-black ones [49]. They also receive less care than they should, because patients with higher medical bills (typically white) are prioritized, since the algorithm deployed uses healthcare cost as a proxy for health, assuming that higher cost means more

severe health problems [50]. Unbalanced datasets lead to higher probability of inaccurate skin cancer diagnoses for people with darker skin colour [51]. Ferryman and Pitcan offer an extensive qualitative study regarding fairness and data set bias in precision medicine [52]. Additionally, AI algorithms may produce unanticipated results, such as predicting race from medical images [53], which may lead to discrimination.

### 3.4. A Definition for AI Ethics in the Big Data Era

Subsequently, Floridi and Taddeo define data ethics "as a new branch of ethics that studies and evaluates moral problems related to data (including generation, recording, curation, processing, dissemination, sharing and use), algorithms (including artificial intelligence, artificial agents, machine learning and robots) and corresponding practices (including responsible innovation, programming, hacking and professional codes), in order to formulate and support morally good solutions (e.g., right conducts or right values)" [54] (p. 3).

This definition is much more specific than the one that is given by Leslie [11] that was cited in Section 2, in the sense that underlines three main constituents (data, algorithms, and practices) that determine the moral issues that are posed by AI as described in the introduction. In this section we briefly addressed the first two (data and algorithms), while the following sections we are going to cover the last one.

## 4. Guidelines

In a 2014 paper [55], Richards and King addressed, from a legal point of view, the issues emerging from the accumulation of big datasets and their manipulation and call for the development of "Big Data Ethics", according to four high-level principles: privacy, confidentiality, transparency, and identity (the ability of individuals to define who they are).

In order to foster ethical behavior in the AI practice, a series of national and international organizations, initiatives, companies, NGOs, governments, professional associations, etc., have developed a series of guidelines, principles, or codes of conduct. The NGO AlgorithmWatch created a global inventory of such guidelines, which they differentiate into binding agreements, voluntary commitments, and recommendations (legislation is not included in the inventory), comprising of 173 contributions at the time of writing [56]. The earliest publication date is 2010 (some contributions are not dated) and the majority are coming from government and civil society organizations of northwestern Europe and North America.

### 4.1. Main Guideline Concepts and Principles

Recent surveys that attempt to map the AI guidelines landscape identify common principles, issues, or themes that run through them as: transparency, justice, fairness, non-maleficence, responsibility, privacy, beneficence (in the terms of sustainability, well-being and common good), freedom, autonomy, trust, dignity, solidarity, accountability, auditability, safety and security, explainability, human control of technology, promotion of human values, humanity, collaboration, share, and AGI/ASI [57–61].

The number of identified principles or issues varies according to both the number and selection criteria of the documents that are surveyed, as well as the granularity of the definition for each respective principle/issue. For example, Jobin et al. [57] and Zeng et al. [61] include transparency and explainability in the general notion of transparency, while Hagendorff [58] examined them as different issues; Floridi et al. included accountability in explicability [60], while for Hagendorff it was a separate issue [58], and Jobin et al. [57] and Zeng et al. [61] included it in responsibility. Regarding the terminology that is used in the various guidelines, Zeng et al. presents a platform that attempts to semantically link the principles in them [61], while Fjeld et al. offers instructive visualizations of the themes that they identify in their report [59].

From the above, it is evident that, as Leslie observes, AI ethics vocabulary is based on two pillars: bioethics, employing the classical four principles of beneficence, non-maleficence, autonomy, and justice (as introduced by Beauchamp and Childress [62]),

for "safeguarding of individuals in instances where technological practices affect their interests and wellbeing", and the human rights discourse, for "social, political, and legal entitlements" [11] (p. 9).

### 4.2. Guideline Critisism

This normative approach is not beyond criticism. Hagendorff [58] observes a number of omissions in the guidelines about issues: aspects of political abuse; governmental control; misinformation and propaganda; related social issues as isolation and echo chambers; the rights of low level AI workers who do data labeling or content moderation; diversity in the AI industry; private-public partnership in AI research; and the ecological impact of AI (from mining and e-waste regarding the hardware to the carbon footprint of the energy consumed for the algorithm training and deployment). He also notes the lack of discussion about AGI, remarking on the fact that the people who worry about it are not usually computer science experts (who regard it as a remote probability), and the absence of deliberation on philosophical problems such as the trolley problem, attributing it again to the fact that the people behind the guidelines are technically educated and not philosophers.

### 4.3. Guideline Application Issues

Additionally, a point that has to be taken into consideration regarding how people might interpret the guidelines, is the distributed nature of many AI systems that usually involves a variety of practitioners with different geographical, cultural backgrounds, and ways of working, as well as belonging to diverse scientific domains, having access to different resources, infrastructures, and funding, which makes agreeing on a set of general principles difficult [63].

Another common criticism is the lack of actionability, namely, that the principles are too high-level to be immediately useful, do not offer specific practices to apply ethics at each stage of the AI/ML pipeline, and often fail to be actioned in governmental policy [64–66]/An interesting parallel can be found in Hegel's criticism of Kant as referenced by Beauchamp and Childress: " . . . Hegel fittingly criticized Immanuel Kant for developing an "empty formalism" that preached obligation for obligation's sake, without any power to develop an "immanent doctrine of duties". Hegel thought all "content and specification" in a living code of ethics had been replaced by abstractness in Kant's account", and they conclude: "If a principle lacks adequate specificity, it is empty and ineffectual." [67] (p. 28).

However, the application of ethics guidelines, especially by companies, raises concern about a number of issues. One of the most prominent is ethics washing [68], where the adoption of self-regulation by the companies is mainly used as an argument to avoid state regulation and control, even by employing in-house philosophers in specifically founded ethics boards. This is an issue that is also noted by Floridi, who refers to this as ethics lobbying [69], as well as by Rességuier and Rodrigues [70] and Benkler [71], the latter stressing the danger of letting the industry alone to shape the future of AI by writing the rules.

This danger is not overstated, taking into consideration that from the 173 guidelines in the AlgorithmsWatch inventory, 115 are non-enforceable recommendations, while regarding the 41 which come from the private sector, 15 are recommendations and 22 are voluntary commitments leaving only 4 characterized as binding agreements, i.e., having the provision for means to sanction non-compliance.

Related is the practice of making misleading and unsubstantiated claims of upholding ethical standards for self-promotional reasons, analogous to greenwashing [68,69].

Additionally, the fact, that principles are highly abstract and can be interpreted differently by different groups with respect to their interests and cultural backgrounds, while they sometimes contradict each other [64] can be exploited for ethics shopping and ethics dumping [69]. In the first case, that entails "mixing and matching" from existing guidelines in order to justify already taken decisions, and in the latter, such as in trade, importing solutions from countries with not so strict ethical commitments.

This instrumentalization of the ethics vocabulary for self-interest or self-promotional reasons led to the related outcomes of ethics shirking, i.e., systematically and intentionally neglecting ethical work, diminishing its importance, status, and impact [69] and ethics bashing, i.e., the depreciation of ethics and their ability to solve issues that are related to the development and use of AI systems [68].

*4.4. Do Guidelines Realy Matter?*

Finally, there is also some evidence that practitioners do not act differently whether they have knowledge of the guidelines or not. A behavioral ethics study involving 63 software engineering students and 105 professional software developers revealed that the decision-making by the participants that were explicitly instructed to consider the Association for Computing Machinery (ACM) code of ethics, was not statistically significantly different when compared with a control group [72]. This is reinforced by a meta-analysis on defining the antecedents of unethical choice in organizations, revealing that the mere existence of guidelines in an organization does not have any detectable impact on moral choices [73]. The authors speculate that this is due to the guidelines' ubiquity which diminishes their potency, or to the fact that they are considered as a mere facade. However, they stress that the effectiveness of the codes of conduct is highly related to whether they are properly enforced.

Another possible explanation is that ethics guidelines being dictated (and probably enforced) by an external body, are regarded by practitioners as irrelevant to the day-to-day activities. Consequently, they are not applied, rather encouraging the mentality of delegating ethical concerns to others [63].

This raises the question of what other methods or techniques could be used instead of (or in addition to) a code of ethics, in order to foster ethical decision-making.

## 5. Moving on from Principles

*5.1. Application Frameworks*

While most of the guidelines refer to abstract principles as noted by Hagendorff, they provide no, or very few notes on technical implementation: only two of the 22 guidelines documents that he surveyed in [58], namely, the AI principles of the European Commission's High-Level Expert Group on AI draft for ethical AI [74] and AI4People—An Ethical Framework for a Good AI Society [60], contain notes on technical implementation. There are a number of approaches that move beyond the declaration of principles into how these can be put into practice, encompassing the already existing technical solutions. These approaches usually consist of defining a framework within which the ethical principles or values are operationalized with regard to the various stages of an AI project process model—usually CRISP-DM (cross industry standard process for Data Mining) or similar. CRISP-DM defines six high-level phases: business understanding, data understanding, data preparation, modeling, evaluation, and deployment [75].

Saltz and Dewar propose a framework that explores the key ethical themes regarding project initiation and management, data, and model challenges in each one of these phases. The ethical considerations that they propose for examination are: personal/group harm, team accountability, data misuse, privacy and accuracy, model subjective design, and misuse/misinterpretation [37]. Their work, inspired Morley et al., in creating an applied AI ethics typology, where they identify tools and methods that can be applied from the bibliography [65]. These are classified according to both each of the five principles that were identified by Floridi et al. [60]: beneficence, non-maleficence, autonomy, justice, and explicability; and the seven stages of AI application lifecycle as presented in the overview of the UK Information Commissioner's Office (ICO) auditing framework for Artificial Intelligence and its core components: business and use-case development, design phase, training and test data procurement, building, testing, deployment, and monitoring [76].

The Alan Turing Institute, in its guide for the responsible design and implementation of AI systems in the public sector, proposes an ethical platform for the responsible delivery



of AI projects, the purpose of which is to safeguard and enable the justifiability of both the AI project and its product. The platform consists of three building blocks: the value: respect, connect, care, and protect; the principles: fairness, accountability, sustainability, and transparency; and a process-based governance framework across the AI system design and implementation workflow processes (according to CRISP-DM but also applicable in other related workflow models) [11].

The High-Level Expert Group on AI (AI HLEG) set up by the European Commission proposes a framework, setting the Ethics guidelines for the realization of trustworthy AI, a concept that comprises lawful, ethical, and robust (i.e., secure and reliable) AI. Trustworthy AI is based on a set of initial principles, namely respect for human autonomy, prevention of harm, fairness, and explicability. These principles, in turn, define seven requirements: (1) human agency and oversight; (2) technical robustness and safety; (3) privacy and data governance; (4) transparency; (5) diversity, non-discrimination, and fairness; (6) environmental and societal well-being; and (7) accountability, which are subsequently implemented by technical and non-technical methods. The technical methods comprise of: system architectures which encompass procedures and/or constraints on procedures which implement trustworthy AI; ethics and rule of law by design, i.e., methods to ensure that abstract principles are translated into specific implementation decisions from the very start; explanation methods; testing and validating; and finally, quality of service indicators, such as measures to evaluate the testing and training of algorithms, functionality, performance, usability, reliability, security, and maintainability. The outcome for each one of the requirements should be continuously assessed during a system's life cycle [77].

The report acknowledges that, while some of these technical solutions still require more research, quite a few are already available today. There is extensive existing research about such technical solutions tackling specific issues that have been developed independently of any framework.

### 5.2. Technical Approaches to Spefic Issues

Since the beginning of the "Big Data" era, data collection, maintenance, and sharing have received specific attention regarding governance [78] and consent [79], while according to the "garbage in-garbage out" maxim, much attention has been given to detecting and mitigating bias in algorithms and datasets [80,81]. Indicatively, the proposed solutions include: data documentation via the introduction of datasheets for datasets in analogy with what exists in the electronics industry; documenting the creation motivation, composition, collection processes, preprocessing/cleaning/labeling, potential uses, distribution, and maintenance of a dataset [82]; using trusted third parties to hold sensitive data, identify data bias beforehand, or act as a pre-processor and giving access accordingly, e.g., by enforcing anonymization [83]; and the use of mechanisms such as data trusts, i.e., legal structures that provide mechanisms "for individuals to pool their data rights (or data) into an organisation—the trust—which is then tasked with governing its use according to conditions stipulated at its establishment" [84] (p. 8). All the above aim at ensuring principles such as safety and security, privacy, fairness, and accountability, which are among the most commonly mentioned in the guidelines [58,59].

Another commonly mentioned principle, namely explainability/transparency (also linked with accountability), is addressed by a substantial body of work in eXplainable AI (XAI). XAI aims to produce explainable models or effective individual explanations, thus enabling users to understand and trust the system (for a comprehensive overview, see [85]). While XAI is often connected to EU legislation -GDPR and "the right to explanation" [86], whether this includes a right to explanation from automated procedures is a debated issue [87].

Security and privacy are two issues that have been extensively researched since the "Big Data" data era with techniques such as anonymization and encryption routinely applied; a glossary of relevant terms as well as an overview of current and next-generation methods for federated, secure, and privacy-preserving artificial intelligence can be found in [88]. The authors define secure AI as the combinations of methods that are concerned

with safeguarding algorithms, while privacy-preserving AI is when the system allows the processing of data without revealing the data itself. The two of them ensure data and algorithm sovereignty, while enabling trustworthiness and transparency, as well as integral computational processes and results. While the authors focus on medical imaging applications, the methods that are described can be widely applied.

A comprehensive review of the state of the art in privacy issues and solutions for machine learning is presented by Liu et al. [89]. AI is examined here as either the object of privacy concerns or the tool to avert or instigate privacy attacks. The authors offer a comprehensive survey on AI/ML and privacy, addressing all three cases and identifying the key challenges. In the first case, where the objective is to ensure the privacy of either the training data or the algorithm, they classify the attacks and protection schemes and assess the existing solutions. Regarding the other two cases, namely AI aided privacy protection and AI-based privacy attack, they conclude that while the first one is gaining momentum, the research about the latter is in infancy but it is expected to have great future development.

## 6. Discussion: A Case of (for) Responsibility

Apart from the technical solutions, the above-described frameworks include or imply non-technical ones. The AI HLEG report identifies the following: regulation, codes of conduct, standardization, certifications, accountability via governance frameworks, education and awareness to foster an ethical mind-set, stakeholder participation and social dialogue, diversity, and inclusive design teams [77]. We have examined codes of conduct in Section 3 (Guidelines), while we consider mandatory approaches such as regulation, standardization, and certification as being outside the scope of this paper. In this section we are going to focus on the rest which we examine under the umbrella terms of responsibility and accountability.

### 6.1. Responsibility vs. Accountability

Responsibility and accountability are sometimes used interchangeably and rarely defined, while there is a dispute whether an AI system can be held accountable or it is always the humans who carry the ultimate responsibility [57].

Here we adopt the definition of responsibility by Leonelli, as "the moral obligation to ensure that a particular task is adequately performed, which is typically associated with someone's social position, function or role and does not necessarily entail being legally or otherwise answerable for one's actions", distinguished from accountability which is defined as "the duty to justify a given action to others and be answerable for the results of that action after it has been performed" [63] (p. 3), not forgetting Hao's tongue in cheek definition of accountability as: "accountability (n)—The act of holding someone else responsible for the consequences when your AI system fails" [90]).

### 6.2. Personal Responsibility

Responsibility can be either focused on the individual AI practitioner as a moral agent but also related to sustainability defined as a "broader objective of promoting mutually beneficial interaction between data science and society" [91] (p. 1). Sustainability is also one of the core principles that is defined by Leslie in [11], as AI systems may have long-term impacts on the communities they are applied to. Additionally, the nature of the AI pipeline makes us take into consideration what is defined by Floridi [92] as distributed responsibility, where human, artificial, and hybrid agents perform morally loaded but neutral actions which, however, in a networked environment, may cause morally loaded results.

Personal responsibility can be assigned to AI practitioners at each stage (according CRISP-DM or an equivalent model) of the AI project; especially data understanding and preparation, modeling, and evaluation stages, where they have greater control, as opposed to the data collection and storing as well as the deployment of the final product and its broader (e.g., societal or environmental) impact. In this framework, Rochel and Evéquoz

provide recommendations mainly focusing on the AI engineer's ability to justify their choice at each stage of the project and communicate the information that is available to them both to the project leaders and the clients, explaining the tradeoffs they had to take into consideration and the possible impact of any decision that is taken [93].

Similarly, Leslie proposes a Discriminatory Non-Harm Self-Assessment for the equivalent stages. Here, self-assessment is related to fairness as a preemptive way of identifying biases leading to discrimination, thus can be considered as falling under the responsibility definition that was adopted by Leonelli in Section 6.1. Accountability on the other hand, is composed of answerability and auditability, the latter answering the question of how AI engineers are to be held accountable for the system outcomes. Thus, auditability in Leslie is more akin to accountability as defined by Leonelli above. Decision justification is defined as answerability, and should run through the whole AI project delivery workflow, establishing a continuous chain to address the distributed responsibility issue [11].

### 6.3. Sustainability

Responsibility is regarded as largely synonymous with sustainability and viewed as a data governance issue in [91]. Specifically, data are seen as a common pool resource (CPR), since it exhibits the characteristics of a complex resource ecosystem, comprising infrastructures, individuals, groups, institutions, and the relationships to each other, which produce, consume and are affected by data.

However, data ecosystems present difficulties with what is considered effective commons governance/ Specifically, according to Dietz [94], as cited by Taylor and Purtova in [91] (p. 3) "effective commons governance is easier to achieve when the following conditions are present: (i) the resource and its use can be monitored at low cost; (ii) the resources, users, technology, and economic and social conditions are changing at a moderate rate; (iii) frequent communication between stakeholders that facilitates trust, lowering the cost of monitoring; (iv) outsiders can be excluded from the resource at low cost; and (v) users support effective monitoring and rule enforcement". As the authors observe, in the case of data commons, the conditions are usually not met, since data processing is ubiquitous, technological change is extremely fast, cross-stakeholder conversation about data is practically not supported, the cost of exclusion from data is prohibitively high for almost everybody, and finally, only a threat of enforcement action and/or major scandals may oblige the key actors to submit information about their data practices. The authors focus on stakeholdership identification, namely who is affected by the AI system, since many people are either unaware that their data are being processed or that they are being used in other ways that they were originally asked (and given consent) for. The authors emphasize the role of institutions, either local, national, or even global depending on the application.

Leslie, on the other hand, addresses the problem by proposing a Stakeholder Impact Assessment for each AI project (delivering a public service or in a back-office administrative capacity), which will be carried out starting from the problem formulation, to pre-implementation and finally, to reassessment after the system has been deployed [11].

The consideration of the well-being of the stakeholders as a constituent of a benevolent organizational ethical climate, i.e., beliefs about what constitutes right behaviour in an organization, is correlated negatively with unethical choice as opposed to an egotistical ethical climate that rewards performance without any consideration on the means used [73].

### 6.4. Responsibility and Accountability in the AI Application in Healthcare

According to Topol, the application of AI in healthcare positively affected medicine regarding "three levels: for clinicians, predominantly via rapid, accurate image interpretation; for health systems, by improving workflow and the potential for reducing medical errors; and for patients, by enabling them to process their own data to promote health." [30] (p. 44). However, AI-assisted decision-making has posed a series of responsibility and accountability issues. The World Health Organization in [95] identifies them as (a) the "control

problem", i.e., the lack of accountability of the AI developers for the possible errors that are committed by their software, placing the burden on the health care professionals who did not participate in its implementation; (b) the distributed responsibility that evolves from the implication of many people in the development of AI applications, making difficult the "traceability of harm"; and (c) the failing of imposing or checking the application of ethical guidelines on the software companies that develop AI.

In the case of autonomous decision-making, even graver problems appear, such as the lack of explainability, in the case of disagreement between the doctor and the AI, due to the black-box nature of most of these systems. This also includes not only the specifics of the algorithm that is used but the absence of information on the training data which may result in problems caused, for example by contextual bias. Contextual bias emerges because Medical AI is usually trained in high-resource settings, such as academic medical centres or state-of-the-art hospital systems but applied in low-resource settings, such as small hospitals or practitioners' offices [96]. The lack of transparency might also affect the process of getting a patient's meaningful consent and jeopardize the patient-doctor relationship. Additional problems include the resource allocation and prioritization, and the use of AI for predictive analytics (as for the cases of assigning appointments and using cost as a proxy for predicting health problems that are mentioned in Section 3). Failing to assign responsibility in all of these cases, results in avoiding holding anyone accountable and undermining trust.

*6.5. Suggestions to Assign Responsibility and Ensure Accountability via Regulation*

Specifically for systems that provide services to the community, for example, criminal justice, medicine, and education, it is advocated that firms should be held responsible (and accountable) for the algorithms they create and sell. The primary reason is that they are the ones (and usually the only ones, since the algorithm is an industrial secret) that are knowledgeable on how the algorithm is designed and implemented, and secondly, the values they incorporate in them by design must respect the norms of the said community.

Additionally, since AI systems are used for decisions regarding access to social goods and impact citizen's rights, AI practitioners should be required to be certified as doctors, civil engineers, and lawyers are, and attend ethics education during their studies [97].

In the same vein, Garzcarek and Steuer [98] as well as Mittelstadt [99] support defining data scientists as a separate profession. The latter proposes licensing for AI practitioners involved in high risk projects and compares AI development to medicine, concluding that the former lacks: "(1) common aims and fiduciary duties, (2) professional history and norms, (3) proven methods to translate principles into practice, and (4) robust legal and professional accountability mechanisms" [99] (p. 2). However, this is something to be expected from a fairly new domain such as AI, especially when compared with disciplines such as medicine (the Hippocratic oath dates from 400 BCE [100]). Additionally, what constitutes high risk is something that is not easily defined; any AI application can have potentially moral implications not intended or (indeed) foreseen by its designers.

*6.6. Beyond Regulation I—Organisational Culture*

The latter may be regarded as "bad cases" as defined by Kish-Gephart, Harrison, and Treviño [73], i.e., moral issues that may provoke or prevent unethical choices by an individual when faced with a moral dilemma. The authors identify the characteristics of a "bad case" by adopting the concept of moral intensity that was proposed by Jones [101] (p. 372) as "a construct that captures of the extent of issue-related moral imperative in a situation". The results or their meta-analysis reveal a high-intensity moral issue is identified by the correlations among the magnitude of consequences, concentration, and probability of effect, and temporal immediacy. That makes a moral issue more prominent in the eyes of a moral agent as having consequences for others. Consequently, if it is possible to highlight these characteristics in organizational decision-making, thus creating awareness that a decision can be associated with severe, well-defined harm, it might be

possible to suppress unethical choice. Organizations can support this process by creating clear behavioral norms. Their analysis shows that a principled organizational climate that encourages employees following rules that protect the company and others, especially when combined with an ethical culture (an organization's systems, procedures, and practices for guiding and supporting ethical behavior [73]) that clearly communicates the range of acceptable and unacceptable behaviors through leader role-modeling, reward systems, and rigorous code enforcement, is associated with fewer unethical decisions. Organizations that fail to do so are identified by the authors as "bad barrels".

### 6.7. Beyond Regulation II—Activism in the Workplace

This is a case where AI community activism could make a difference. The AI community includes, according to Belfield, "researchers, research engineers, faculty, graduate students, NGO workers, campaigners, and some technology workers more generally— those who would self-describe as working 'on', 'with', and 'in' AI and those analyzing or campaigning on the effects of AI" [102] (p. 15). The author examines the achievements and future prospects of said activism with respect to two frameworks: epistemic communities and worker organizing. In both cases, the AI community has achieved some success owing, on one hand, to the high educational and societal status of its members, and on the other, to their bargaining power caused by talent scarcity and high demand. However, as the author points out, the future of activism depends on two factors: the balance of talent supply and demand and the cohesiveness of the AI community. High cohesiveness due to the shared education and work experience, can be weakened by the need of broadening the recruitment pool, owing both to talent scarcity and expertise differentiation. With further development of AI developer tools, not everybody needs to be highly skilled (either by education and/or experience) and consequently are lower compensated and more replaceable. The AI community is constituted not only of AI engineers but also of gig workers, such as the ones that are employed by Amazon Mechanical Turk ("Turkers"), who definitely work 'on', 'with', and 'in' AI; are not highly paid [103]; while producing potentially highly impact work, for example in the formation of golden standard datasets [104].

### 6.8. Beyond Regulation III—Education

Education on ethics can be a way of raising awareness and identifying potential morally critical situations, and is advocated for the AI practitioners as a part of their formal education to be included in university curricula [63,98,105,106], for the general population even starting from school as part of a digital literacy course [107], or both [60,108]. Eaton et al., support both the teaching of AI experts and students in non-computing subjects to have a basic understanding of AI techniques [109] while Moor supports a solution through greater incorporation of politics in the ethics curricula in computer science [110]. However, as Kish-Gephart, Harrison, and Treviño show in their meta-analysis, that when excluding personal psychological characteristics, certain demographics (gender, age, and education) are inconsequential regarding unethical behavior [73]. Education here means formal education, the hypothesis being that higher education supports general cognitive and social development, a feeling of greater personal responsibility and probably some ethics training as well. The authors pose the unsettling question of whether higher education is abdicating its responsibility regarding the ethical development of students, given the fact there exist so many prominent cases of highly educated persons that fail to reach ethical decisions.

However, they conclude that unethical choices have multiple origins: certain types of individuals that are encouraged by a specific organizational culture make unethical decisions when faced with moral dilemmas. Consequently, there is no silver bullet; they conclude suggesting a combination of selection, training, and management practices.

## 7. Conclusions

The rise and wide application of AI in recent years has impacted the lives of an increasing part of the population and raised a variety of ethical issues mainly about the collection, management, processing of data, opacity of algorithms, and the deployment of results.

Issues such as bias, either in the data collection and preprocessing stages or in algorithm design, leading to discrimination against vulnerable populations in critical areas, such as law enforcement, healthcare, labor market, and financial services, as well as issues of privacy, security, and accountability, have led to the deployment of a variety of actions, legal, technical, and normative towards ethical or responsible AI.

We have broadly sketched the landscape of AI ethics, starting from a historical perspective and focused on data-driven, non-symbolic AI, the related moral issues and ways to deal with them, mainly through guidelines, principles, and codes of conduct and their application via frameworks, suggesting technical and non-technical solutions. Finally, we dealt with the concept of responsibility, its meaning within the context of personal responsibility and accountability, as well as sustainability, namely taking into consideration social beneficence.

We examined unethical behavior through the prism of personal characteristics, high moral intensity issues, and organizational culture, coming to the conclusion that unethical choice is a multifaceted phenomenon, so it must be addressed by a variety of means, including legislation, guidelines, and education

AI ethics is a multi- and inter-disciplinary domain that is related to computer science, business ethics, law, philosophy, and psychology. While there is currently a lot of activity mainly around legislation and guidelines, there is fertile ground for debate involving multiple actors: the AI community, organizations and companies, civil society, governments, and intergovernmental blocs.

Education and general awareness raising both for AI practitioners and the general population is an obvious path to follow, putting our faith in the Socratic apostrophe in Protagoras:

"That whoever learns what is good and what is bad will never be swayed by anything to act otherwise than as knowledge bids, and that intelligence is a sufficient succor for mankind?" [111] (section 352c).

As a final note, we wish to point out the possibilities of automated moral decisions. Since AI systems are far more complicated and faster than any human can cope with, many ethically-loaded decisions (especially if they need be to be conducted in real-time) may be delegated to automation, following the pattern of social media moderator algorithms. While this can be prohibited for high-impact and sensitive systems, it might also be the case that it cannot be avoided. Consequently, questions such as whether ethics is computable, posed by Moor [17], if an artificial system can be a full moral agent, whether only humans can be held accountable, and distributed responsibility, could be put on the spotlight for further research.

**Author Contributions:** All authors have contributed equally to the creation of this article. All authors have read and agreed to the published version of the manuscript.

**Funding:** This research was partially funded by the German Federal Ministry of Education and Research (BMBF) FAIR Data Spaces project by grant number FAIRDS15.

**Institutional Review Board Statement:** Not applicable.

**Informed Consent Statement:** Not applicable.

**Data Availability Statement:** Not applicable.

**Conflicts of Interest:** The authors declare no conflict of interest.

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
