# Peer review of "AI Ethics—A Bird’s Eye View"

_applsci, doi:10.3390/app12094130_

Round 1
Reviewer 1 Report
This paper, entitled "AI Ethics - a bird's eye view", is a review of past and current research efforts in AI ethics.
The paper is of interest to AI professionals looking to follow ethical guidelines in their activity, as well as researchers of this field.
The paper presents (1) the definitions and accompanying concepts related to AI ethics, (2) the ethical issues related to collecting/processing/analysing datasets, (3) existing guidelines for the ethical use of AI, (4) existing legal frameworks, (5) existing views on assignment of responsibility and accountability regarding the use of AI systems, as well as a discussion on this topic.
The paper is clearly written, but the thematic differences between its sections are subtle. The bibliographical review is satisfactory, yet the novelty of this review is not evidently stated, as compared to older reviews (e.g. "From What to How: An Initial Review of Publicly Available AI Ethics Tools, Methods and Research to Translate Principles into Practices"). The discussion is impartial, raising relevant criticisms of dishonest practices in the industry related to AI ethics.
The quality of English is good, although with multiple typos.
Please find below my detailed comments:
Literature review:
- Line 110: Please provide some self-sufficient and concise explanations and references to the philosophical approaches of "consequentialism", and "Kantianism".
- Lines 189-198 can be compacted to remove duplicate mentions of common principles.
- Line 198: What is ASI ? Please provide an explanation for the acronym.
- On page 10, the footnote 12 is somewhat cryptic. The "Obviously" stated phrase is not obvious.
Suggestions:
- The administrative affiliation (lines 4 and 6) can be separated from your emails (lines 5 and 7), to avoid duplication. Emails are firstname.lastname@uk-koeln.de.
- Footnote 17 does not bring any value to the text, and can be safely removed.
Minor typos:
- Line 40: not only (written twice)
- Line 52: Boyd.
- Line 115: "the there"
- Line 139: separating the phrase into multiple ones would improve readability.
- Line 172: "the first two" (does not link with the rest of the phrase)
- Line 185: "is excluded" (does not link with the rest of the phrase)
- Line 186: "earlier" -> "earliest" ?
- Line 310: missing word.
- Line 317: "are and"
- Line 402: "focus" -> "focusing"
- Lines 454-455: "more prominent [...] as having consequences for others" -> the phrase is not clear.
- Line 482: "highly impact work" -> "high impact work".
Author Response
Dear reviewer,
Thank you for your time and effort in reading, and assessing our paper, as well as providing us with detailed, insightful, and helpful remarks and suggestions that will considerably improve its quality.
Regarding your general remarks, the main purpose of the paper was to provide an introduction to AI ethics to the audience of a special Issue on Data Science for Medical Informatics. We assumed that while a lot of people might be interested or even familiar with some issues relevant to their work (e.g. bias), might not be aware of the multifaceted nature of the discipline. Thus we tried to provide some general overview and give pointers for further interested readers in the references. Our aim was that the reader would be able to see AI ethics issues with in a variety of views, philosophical, legal, societal and lead to a discussion regarding the assignment of responsibility and accountability and how these relate to the individual and the community (in the form of state regulation, education, civil society, business ethics, etc.). While these aspects can be found in many of the papers reviewed, we thought that the reader would benefit from a broader (albeit necessarily coarser) perceptive, a “bird’s eye view”.
Regarding your specific remarks, please see the attached file

Reviewer 2 Report
The paper presents a real issue of AI ethics. The paper is good, but minor spell check is required (for example repetition in line 40).
The article is difficult to follow and read if the reader does not have enough background in Ethics, law, etc. The authors can improve section 3 by providing the issue (with a title/name) then discussing it. This also can be done in section4. Also, the suggestions given in Section 6 can be summarized, at the end, in brief points.
Author Response
Dear reviewer,
Thank you for your time and effort in reading, and assessing our paper, as well as providing us with helpful remarks and suggestions that will considerably improve its quality.
Regarding the background the reader needs in order to follow the paper, we recognize that it spans in many areas outside the immediate research interests of the readers, but it is due to the multidimensionality of the issue.
We try to provide the reader with explanatory information in the footnotes and points to the bibliography for further study. We hope that this helps considerably for the comprehension of the paper.
Regarding your specific remarks, please see the following table
|
Point no |
Comment |
Response |
|
1 |
The authors can improve section 3 by providing the issue (with a title/name) then discussing it. |
Done |
|
2 |
This also can be done in section4 |
Done |
|
3 |
Also, the suggestions given in Section 6 can be summarized, at the end, in brief points. |
We have also inserted subsections. Their titles serve also as summary points. We chose that in order to avoid repetition. Also, most of the suggestions are also summarized in the conclusions |
Reviewer 3 Report
The article that is presented is a theoretical review that does not present a methodology of this type of work.
Author Response
Dear reviewer,
We would like to thank you for taking the time to read and assess our paper.
Regarding the point that you make that this review is theoretical and does not follow the usual methodology for this kind of reviews, we would like to answer that it was not mend to be an exhaustive review of the area, rather an introductory one.
Since it is aimed at a special Issue on Data Science for Medical Informatics, we intended to provide a bird’s eye view (hence the title), to AI ethics, a topic relevant to Medical Informatics but not widely familiar. Our purpose was to sketch in broad strokes the AI ethics landscape, as to provide some general overview and give pointers for further interested readers in the references. Thus, we chose to include selective bibliography describing the main concepts and underlining the basic issues and challenges in the domain. We followed a two dimensional approach, first a “historical” one by describing the general AI ethics issues which emerged from the beginning of the AI endeavour to today’s Data driven AL/ML . The second dimension addresses the multidisciplinary nature of AI ethics, a domain that involves philosophy, computer science, business ethics, psychology, sociology and is related to law via the call for regulation.
We hope that the above clarification might offer another view to reassess the paper under a new light.
Reviewer 4 Report
This submission to Applied Sciences offers a fascinating glimpse into the landscape of AI ethics through an emphasis on the concept of responsibility. The analysis consists of a critical review of current literature in the areas of AI ethics, data ethics, and big data. Adopting a "bird's eye" perspective, the article offers an accessible account, especially for readers who are new to AI ethical debates.
To strengthen the analysis, however, I recommend a few modifications, as outlined below. While the "bird's eye" view of the AI landscape is valuable, the submission suffers from a general lack of coherence, focus, and argumentation.
(1) The principal ethical issues related to AI are mentioned (environmental, social, cultural, and so forth) in a diffuse manner but not explored in depth. A case study, for example, demonstrating AI impacts on the environment or AI implications in racial profiling would help to focus the article. This will allow readers to take a ground-level view to balance to broad emphasis.
(2) The emerging issue of automated moral decisions is mentioned at the very end of the article (Section 5 Conclusions) but would be more valuable if foregrounded earlier in the discussion and explored in more depth, perhaps even through a focused case study.
(3) The argument of the article needs to be clarified early on. What is the major finding resulting from your analysis of the literature? Is it that AI is "a multi- and interdisciplinary domain" and thus necessitates a broad-based approach? How might such an approach be implemented at various levels to ensure ethical frameworks are developed and taken seriously for AI practitioners?
(4) The environmental implications of AI and big data could be further articulated, i.e. in terms of waste generation and other concerns. This will help to add depth to the analysis.
(5) More care needs to be taken with the presentation of the article. Avoid run-on sentences and non-grammatical constructions. Rather than citing the numbered reference parenthetically (as for example [5]) give the author names with the numbered reference at the end of the sentence where possible. Make sure the revised version is proofread.
Author Response
Dear reviewer,
Thank you for your time and effort in reading, and assessing our paper, as well as providing us with helpful remarks and suggestions that will considerably improve its quality.
Regarding the specific points made :
Points (1), (2) and (4). The paper was conceived as a broad introductory review of AI ethics, in order to give insights of the area from multiple domain view points to the readers of the special issue, while providing bibliographic pointers for specific subjects. Thus, it is unavoidable that we had to deal in a synoptic manner with a multiplicity of very interesting and important issues. Thus, we chose not to extensively present case studies, as that would make the paper much longer and complicated and it might dissuade some readers in engaging with it. However, your remark encouraged us to add specific sections on health care AI applications through the paper (specifically, about discrimination, the use of automation and the assignment of responsibility) with the respective references for the interested reader. Since this is a special issue on Medical Informatics, we thought that this approach will be more interesting to the readers.
Regarding point (3), we added a section in the introduction, which we hope clarifies the objectives
Regarding (5) we tried to follow your suggestions with respect to the English language (to the extent of our abilities). We gave the paper to interested colleagues for proof reading and hopefully the number of typos and errors is considerable less than the first version